# Cross-Linking Mass Spectrometry on P-Glycoprotein

**DOI:** 10.3390/ijms241310627

**Published:** 2023-06-25

**Authors:** Gabriella Gellen, Eva Klement, Kipchumba Biwott, Gitta Schlosser, Gergő Kalló, Éva Csősz, Katalin F. Medzihradszky, Zsolt Bacso

**Affiliations:** 1MTA-ELTE Lendület Ion Mobility Mass Spectrometry Research Group, Department of Analytical Chemistry, Institute of Chemistry, ELTE Eötvös Loránd University, H-1117 Budapest, Hungary; gabgellen@staff.elte.hu (G.G.); gitta.schlosser@ttk.elte.hu (G.S.); 2Department of Biophysics and Cell Biology, Faculty of Medicine, University of Debrecen, Egyetem tér 1., H-4032 Debrecen, Hungary; kipchumba.biwott@med.unideb.hu; 3Doctoral School of Molecular Cell and Immune Biology, University of Debrecen, Egyetem tér 1., H-4032 Debrecen, Hungary; 4Single Cell Omics Advanced Core Facility, HCEMM, H-6728 Szeged, Hungary; klement.eva@brc.hu; 5Laboratory of Proteomics Research, BRC, H-6726 Szeged, Hungary; medzihradszky.katalin@brc.hu; 6Department of Biochemistry and Molecular Biology, Faculty of Medicine, University of Debrecen, Egyetem tér 1., H-4032 Debrecen, Hungary; kallo.gergo@med.unideb.hu (G.K.); cseva@med.unideb.hu (É.C.); 7Proteomics Core Facility, Department of Biochemistry and Molecular Biology, Faculty of Medicine, University of Debrecen, Egyetem tér 1., H-4032 Debrecen, Hungary; 8Faculty of Pharmacology, University of Debrecen, Egyetem tér 1., H-4032 Debrecen, Hungary

**Keywords:** P-glycoprotein, cholesterol, cancer, multidrug resistance, cross-linking mass spectrometry, mono-link, membrane, protein structure

## Abstract

The ABC transporter P-glycoprotein (Pgp) has been found to be involved in multidrug resistance in tumor cells. Lipids and cholesterol have a pivotal role in Pgp’s conformations; however, it is often difficult to investigate it with conventional structural biology techniques. Here, we applied robust approaches coupled with cross-linking mass spectrometry (XL-MS), where the natural lipid environment remains quasi-intact. Two experimental approaches were carried out using different cross-linkers (i) on living cells, followed by membrane preparation and immunoprecipitation enrichment of Pgp, and (ii) on-bead, subsequent to membrane preparation and immunoprecipitation. Pgp-containing complexes were enriched employing extracellular monoclonal anti-Pgp antibodies on magnetic beads, followed by on-bead enzymatic digestion. The LC-MS/MS results revealed mono-links on Pgp’s solvent-accessible residues, while intraprotein cross-links confirmed a complex interplay between extracellular, transmembrane, and intracellular segments of the protein, of which several have been reported to be connected to cholesterol. Harnessing the MS results and those of molecular docking, we suggest an epitope for the 15D3 cholesterol-dependent mouse monoclonal antibody. Additionally, enriched neighbors of Pgp prove the strong connection of Pgp to the cytoskeleton and other cholesterol-regulated proteins. These findings suggest that XL-MS may be utilized for protein structure and network analyses in such convoluted systems as membrane proteins.

## 1. Introduction

The integral membrane protein P-glycoprotein (Pgp), also known as multi-drug resistance protein 1 (MDR1) or ATP-binding cassette sub-family B member 1 (ABCB1), acts as a guard at barrier regions (e.g., endothelial cells and blood–brain barrier) protecting organs against toxic substrates. However, it can be overexpressed in cancer cells, and the efflux of various lipophilic therapeutic drugs contributes to multidrug resistance (MDR) [1].

Pgp is a member of the ATP-binding cassette (ABC) transporter family based on its structure and the mechanism of its ATP hydrolysis. Pgp consists of 12 transmembrane helices (TMHs) forming 6 extracellular loops (ECL) and 2 transmembrane domains (TMDs). The two TMDs are connected both directly to the intracellular ATP binding domains (nucleotide-binding domains, NBDs) through alpha-helices (bridging TMH6 to NBD1 and TMH12 to NBD2) and indirectly through non-covalent interactions via intracellular coupling helices (ICH1-4) [2,3]. Nucleotide-binding sites (NBS) are formed by intracellular helices and loops, creating two pockets for ATP binding. The two homologous halves of the protein are connected by a highly flexible linker peptide (Figure 1A,B). During its catalytic cycle, Pgp alternates between two main conformations, the substrate-binding inward-facing (IF) and the substrate releasing outward-facing (OF) state. A third intermediate conformation, the pre-hydrolytic (occluded) structure, has also been determined [4], which is supplemented with numerous smaller slow movements between the main conformations [5], which are chiefly influenced by Pgp’s lipid environment.

Since Pgp is embedded in the membrane, crosstalk between the lipid environment and the protein is naturally expected. Lipids [6] and cholesterol [7,8,9] alter Pgp’s structure, and therefore its proper functioning in drug binding and ATP hydrolysis. Cholesterols have been suggested to influence Pgp’s functioning not only indirectly by modifying characteristics of the membrane, e.g., in membrane lipid rafts and caveolae, but also directly, binding to Pgp even in non-raft membrane microdomains [10]. Cholesterol-binding domains (cholesterol recognition amino acid consensus domains: CRAC, CRAC-like, CARC, and CARC-like [11]) have been demonstrated to bind cholesterol in ABCG2 [12]; also, these sequences can be identified on Pgp around the TMHs (Figure 1A) and many of them were experimentally determined on the substrate-bound human Pgp in lipidic nanodiscs based on the cryo-EM PDB structure (6qex) [13].

Although the structure of Pgp and its alterations during its catalytic cycle have been studied extensively, the effect of cholesterols and lipids is often not considered. Reasons for the scarcity of structural studies on membrane proteins, in general, are difficulties with handling and maintaining their native state; furthermore, conventional structural biology approaches (X-ray crystallography or NMR) require the isolation of the protein from its native environment and the removal or partial replacement of lipids.

That is, the lipid environment and cholesterol have a pivotal role in the structural changes of Pgp. Structural movements of the ECLs can be tracked on living cells in physiological conditions by monoclonal antibodies (mAbs) binding from outside the cell. Upon cholesterol modulation, changes could be observed at the ECLs of Pgp [14], followed by utilizing a mouse monoclonal IgG_1_ antibody, the 15D3. As many other extracellularly acting anti-Pgp antibodies (MRK16, 4E3) may compete in binding to the ECLs, so 15D3 is in competition with the conformational sensitive UIC2 monoclonal IgG_2a_, another mouse antibody, binding to the native Pgp only in its OF state [15]. While it is known that the UIC2, which is not cholesterol-sensitive, favors the substrate-binding ATP-free conformation, when the discontinuous epitopes at the first, third, and fourth extracellular loops (ECL1, ECL3 and ECL4) are close to each other [15,16] the binding sites of 15D3 have not been thoroughly investigated yet. Briefly, 15D3 binds to all cell surface Pgps (Pool1 + Pool 2) [14] independent of substrate presence, unlike UIC2, which labels only 10–40% of cell surface Pgps in the absence of substrates (Pool 1), which is complemented with the remaining fraction (Pool 2) of cell surface Pgps upon substrate binding (“UIC2-shift”) [17]. Additionally, 15D3 and UIC2 have overlapping epitopes since in the presence of the Pgp substrate Cyclosporin A (or other substrates), preincubation of the transporter expressing MDR cells with UIC2 decreased the binding affinity of 15D3 [14]. Therefore, it can be assumed that 15D3 could bind ECLs close to the UIC2′s epitope but rather to those loops that stay relatively close together during the whole course of the catalytic cycle and not be altered by the Pgp’s major conformational states.

Our intention was to investigate the molecular structure of Pgp in near-physiological conditions, maintaining its native lipid environment, in order to experimentally determine the exact epitopes of the cholesterol-dependent 15D3 antibody on the extracellular side of Pgp. Our approach to this was cross-linking mass spectrometry (XL-MS), which has become a widely acknowledged supplement to NMR and X-ray crystallography in recent years. By using appropriate cross-linking agents, covalent bonds may be formed between different domains of a protein or protein interacting partners. Therefore, it is possible to detect connected peptide fragments even after stringent sample preparation.

The most frequently used cross-linking agents, the amine-reactive NHS esters, can covalently bind to lysine residues. Usually, surface exposed lysines are often involved in post-translational modifications [18]. That is, these residues, unless occupied, may be more accessible on protein surfaces to interact with, making it even more feasible to target them for cross-linking with functional protein partners. In the Pgp, there are four lysines in the extracellular regions, which contain the epitopes of antibodies. Additionally, there is one lysine in the 15D3 antibody, located in the antigen-binding complementarity-determining regions (CDRs), and four lysines which are in proximity to the CDRs (~20 Å) which may be cross-linked to the antigen carrier protein under appropriate steric conditions. In addition, since cross-linkers can be cell-impermeable or cell-permeable, these properties may be further incorporated into the experimental design to facilitate cross-linking. Following this strategy, the protein vicinity of the membrane proteins in their native lipid environment may be examined, whereas it is inaccessible using electron-microscopic methods.

In vivo cross-linking methods [19], plasma membrane proximity labeling [20] and the combination of mass spectrometric (limited proteolysis-coupled mass spectrometry, LiP-MS and XL-MS) and crystallographic approaches for membrane proteins [21] are under extensive development. However, no standard approach has yet been established for specific cell surface cross-linking.

In this study, facile approaches with XL-MS were tested on Pgp, which could be applied to studying other integral membrane proteins as well. Pgp was cross-linked with commonly used and commercially available lysine-reactive NHS ester cross-linkers of different lengths and solubilities. The reaction was carried out either in vivo on living cells maintaining the natural lipid environment or on-bead, where some essential in situ cholesterols still remained around the protein. Pgp was purified by means of the above-mentioned mAbs, and analyzed using LC-MS/MS. The results revealed cross-links and mono-links in various regions of Pgp, including areas involved in cholesterol-dependent conformational changes. The study also identified mono-links and cross-links in flexible, disordered regions often missing from the crystal structures of Pgp. Our approaches successfully enriched Pgp and associated proteins, demonstrating its potential as a valuable tool for investigating complex proteins in their physiological conditions.

**Figure 1 ijms-24-10627-f001:**
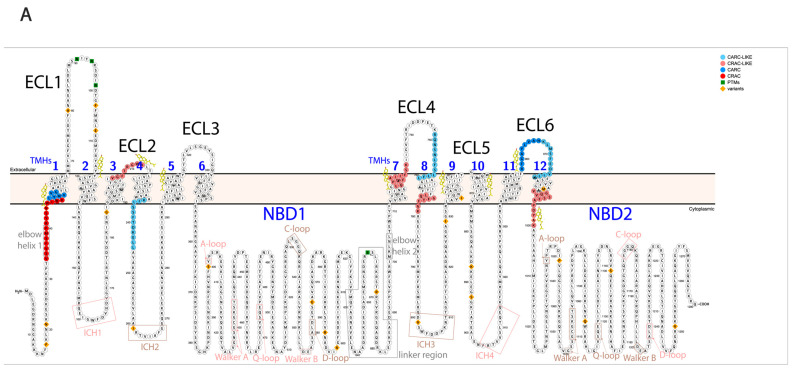
Regions of Pgp on its primary and tertiary structures. (**A**) Primary structure of Pgp visualized using Protter 1.0 [22]. Regions of the protein indicated in pink form nucleotide-binding site 1 (NBS1) and the ones indicated in brown belong to NBS2. TMH 1–12 are indicated in blue. Cholesterols identified in the 6qex PDB structure are labeled in yellow. (**B**) Tertiary structure of Pgp using inward-facing human Pgp structure, 6qex, with its specific regions indicated. TMD1 (light blue) connected to NBD1 (dark blue), TMD2 (pale green) connected to NBD2 (dark green), NBS1 regions (pink), NBS2 (brown) and flexible linker and elbow helices (grey). N-glycans on ECL1 are blue, while oxygen atoms are labeled in red (see abbreviations in the text of the Introduction).

## 2. Results and Discussion

### 2.1. Development of Sample Preparation Approaches for XL-MS Analysis of Integral Membrane Proteins

Cross-linking reactions were carried out either in living cells on which membrane preparation was then performed, or on-bead after membrane preparation. Cross-linked complexes containing Pgp were enriched by means of 15D3 or UIC2 mAbs on Protein G magnetic beads. After stringent washing of the beads, on-bead trypsin digestion was performed prior to LC-MS/MS analysis (Figure 2).

Our approaches were based initially on the rapid immunoprecipitation mass spectrometry of endogenous proteins (RIME) method [23] supplemented with membrane preparation of the cells. Our way of membrane preparation enriches membrane proteins with one or more transmembrane-spanning domains and the connected peripheral and cytosolic proteins in a non-denaturing environment [24]. This approach is more robust and reproducible than other conventional membrane preparation methods are; it also preserves some of the lipids and cholesterol around the protein. Therefore, the tertiary structure of the protein is maintained, just like in antibody binding.

Cross-linking of antibodies to surface proteins in living cells is widely applied when cells are fixed with formaldehyde. However, formaldehyde being a promiscuous cross-linker renders it challenging to identify cross-links [25]. Cross-linkers reacting with specific functional groups, such as primary amines on lysine residues in the case of NHS esters, can help cross-link identification. Previously, anti Pgp mAbs, MRK16 and MRK17 were shown to be cross-linked specifically to Pgp on the surface of living cells by means of a DSP (dithiobis(succinimidyl propionate) NHS cross-linker [26].

We applied the commonly used DSSO and BS2Gd_0_/d_4_ NHS ester cross-linkers which primarily target lysine residues, although serine (12.5% of all cross-linker containing species), tyrosine (4.3%), and threonine (3%) have also been described to be targeted by NHS esters [27,28]. In our experience, however, NHS reagents reacted with these residues only to a negligible extent. Additionally, according to the recommendations of the XL-MS community [29] it is more advantageous to compromise considering only lysines as the reactive site of NHS esters in order to avoid false positive identifications on a complex dataset such as ours. Lysine residues on proteins are usually abundant and readily solvent-accessible; however, integral membrane proteins often lack lysine residues in the transmembrane regions. Thus, employing various other linkers in parallel, for example targeting more acidic amino acids (aspartic acid, glutamic acid), may aid in the mapping of contacting sites of membrane proteins.

Additionally, an extracellular PNGase F (Peptide -*N*-Glycosidase F) treatment was applied on Pgp since it increased antibody binding (Figure 3A,B) by removing all *N*-linked oligosaccharides from Pgp’s ECL1, which altered Pgp’s migration in the electrophoretic field as detected via Western blot analysis (Figure 3C). PNGase F cleaves between the innermost GlcNAc and asparagine residues from *N*-linked glycoproteins [30]. Removal of the N-glycans of Pgp has previously been reported to not influence the function and structure of Pgp [31]. Most probably, glycosylation hampers antibody binding sterically, and removing *N*-linked oligosaccharides allows a more exposed antigen interface. For mapping plasma membrane proteins, this extra step might aid antibody binding in general.

Parallelly to the cross-linking experiments on living cells, on-bead cross-linking approaches were also carried out to target lysines on Pgp more directly. However, in this case, an overabundance of Protein G on the magnetic beads interfered with the identification of other proteins carrying cross-links. In order to reduce cross-linking of lysines on Protein G and also its digestion by trypsin, reductive methylation [32] was performed prior to immunoprecipitation. The derivatization of Protein G greatly diminished its presence in the digests while the binding affinities of the antibodies remained similar to those previously detected. This pretreatment of Protein G enabled a more effective analysis of proteins of interest compared to the previous attempts without methylation.

In our sample preparation approaches (living cell and on-bead), the lipid composition of the obtained protein preparations varies significantly. Using the living cell approach, cells were cross-linked immediately after the culture medium was removed without allowing the lipid environment of the membrane proteins to change. In these samples, the amount of cholesterol in the cell membrane did not change. During the on-bead sample preparation the hydrophobic membrane proteins were first isolated by applying a mild zwitterionic detergent and then, later, were cross-linked. In this case, the lipid profile of the proteins involved may change slightly; e.g., they can lose cholesterol, but at the same time, the immediate advantage is that the amount of the non-membrane proteins in the preparation is significantly reduced. Thus, by applying the two approaches, we can compare the effect of membrane protein structural changes caused by the reduced cholesterol amount or the slightly altered lipid environment with the membrane protein structure that can be expected in an entirely physiological lipid environment under normal cholesterol conditions.

### 2.2. Mono-Links on Pgp in the Native Environment Correlate with Hydrogen/deuterium Exchange Mass Spectrometry (HDX-MS) Results

The application of a cross-linking agent on proteins can result not only in cross-links between specific amino acid residues but the more abundant partially hydrolyzed cross-linkers, since the so-called mono-links are also formed. The probability of forming a mono-link is much higher than that of forming a cross-link. Mono-links, however, give information on solvent accessibility and hydrophobicity, which has been used for modeling protein structures [33].

In samples cross-linked with DSSO on-bead or in living cells or cross-linked with BS2Gd_0_/d_4_ in living cells, altogether 28, 19 and 9 unique mono-links were identified on the intracellular side of Pgp, respectively (Figure 4). On 15D3 and UIC2 mAbs, 10-10 monolinks were identified extracellularly with the on-bead sample preparation approach. The overlap of mono-linked sites on Pgp between different sample preparations is shown in Figure 4. Residues identified in all sample types are K271, K380, K515, K808, K915, and K1150, which indicate that these are the most readily accessible lysine residues of the protein. Residues K433 and K1076, which are in the two opposite Walker A regions of the two NBDs, were found to be mono-linked in the on-bead samples; K1076 was also identified in living cell DSSO experiments. K536 right next to the C-loop constructing NBS2 was mono-linked, and furthermore cross-linked as well (see below). Changes in the structure of these domains upon nucleotide binding are highly investigated [16,34,35]. Thus, identifying mono-links in these regions may support endeavors for analyzing the catalytic cycle of Pgp in living cells.

The same regions of Pgp, where mono-links were identified in this study, could be deuterated in previous HDX-MS investigations [4,5,8]. One exception is lysine K149, which was mono-linked in both on-bead and living cell DSSO experiments. Additionally, K149 was in a cross-link with K146 in the on-bead DSSO samples (see below). The high degree of agreement between HDX results and our XL-MS results substantiates that mono-links indeed indicate solvent accessibility. Moreover, regions of some modified residues (K411, K550, K1093, K1099, and K1212) have been reported to be modulated upon changing cholesterol levels in the membrane from HDX experiments [8]. Additionally, mono-linked K1002, connected to ECL6 is in proximity to a CRAC-like domain, where indeed a cholesterol was identified with cryo-EM (Figure 1A,B) [13]. Although these previous experiments (HDX and crystallography) provided new insights into the structure of Pgp, they either lack exact connections and correlations between motions of residues or miss flexible regions and dynamics of the protein. Additionally, they are often carried out on recombinant mouse Pgp instead of human Pgp, which is more difficult to isolate while maintaining its proper folding, and yet is incorporated in nanodiscs; it might not be a close enough model of the physiological lipid conditions. That is, in our experiments, we used much more robust methods compared to HDX or crystallography, keeping a near-physiological milieu for human Pgp, and it was still possible to detect modified residues that can be beacons of structural changes upon modulations in the lipid environment.

### 2.3. Cross-Links on Pgp Reveal Connections between Opposite TMDs and NBDs

The less abundant cross-links carry information about the solvent accessibility of the two linked residues and the distance constraint between them. DSSO and BS2G are among the most frequently used cleavable and non-cleavable cross-linkers [36]. Using collision-induced dissociation (CID), C–S bonds next to the sulfoxide in DSSO are cleaved at lower fragmentation energies compared to those used to cleave the peptide backbone and this results in characteristic signature ions in the MS2 spectra [37]. Using these distinct patterns, it is possible to recognize cross-links and mono-links automatically upon measurement and data analysis.

Non-cleavable cross-links are more challenging to detect and analyze since the number of possible cross-linked peptides increases quadratically with the number of peptides in the database (n^2^ problem) [38]. In this study, we used a 1:1 ratio of BS2Gd_0_ to deuterated BS2Gd_4_, which allowed the more precise fragment ion matching of modified peptides utilizing a characteristic shift between unlabeled and labeled peptide spectra [39].

The two types of cross-linkers we used have different hydrophobicities and spacer arm lengths. While BS2G is readily water soluble with a linker of 7.7 Å, DSSO is more membrane-permeable, and the spacer’s length is 10.3 Å. These differences between the attributes of the cross-linkers can be one explanation for the diversity of results between the different sample preparation approaches.

Based on solvent accessibility prediction by WESA [40], among the 85 lysines in Pgp, 70 are surface exposed, and 15 are buried, from which it was possible to chemically label 34 (49%) and 5 (33%), respectively, with the above-mentioned cross-linkers. Among all lysines, 39 (46%) could be labeled in all our experimental approaches. Buried lysines are mainly located close to the membrane both intra and extracellularly (Figure 5A), and the transmembrane region is water-inaccessible (Figure 5B). Labeling of the rather buried lysines around the inner cavity (K826) means cross-linkers may enter the protein similarly to Pgp substrates creating covalent bonds there as well.

Most of the cross-link identifications were found in the on-bead samples with 10 unique cross-links on Pgp, while the living cell DSSO samples contained 4 cross-links, and in the BS2Gd0/d4 samples, no cross-link could be detected (Figure 6A,B). Extracellularly, one cross-link was identified on the constant heavy chain of UIC2 mAb. No cross-links were identified unambiguously between Pgp and the antibodies, and neither of them had cross-linked interactions with other protein neighbors of Pgp (inter-links). Furthermore, no inter-protein cross-link could be found between different Pgp molecules, which was corroborated via Western blot analysis as well, where no Pgp dimer formation could be detected (Appendix A). Therefore, all cross-links detailed in this study refer to intra-protein cross-links (intra-links) of Pgp or the antibodies. The quantity of the identified cross-links from the living cell or on-bead approaches is at the same order of magnitude as that previously reported in similar studies [41,42]. Membrane proteins being difficult to handle, the number of identified cross-links in the present study is considered efficient enough and some functional conclusions could be drawn. All cross-links in the living cell DSSO samples were found in the on-bead samples except for one, the cross-link between K279 and K786. Changes in the structure of Pgp in the two types of samples can bring about these discrepancies.

The length of the cross-links was accepted to be between 5 Å and 30 Å based on previously published observations [43,44]; in this way, all identified cross-links could fit on the 6qex PDB structure (Figure 6C and Appendix A). These permissive distance limits are due to the high flexibility of lysine residues and motile intrinsically disordered regions (IDR) of proteins.

IDR loops are often challenging to maintain during crystallographic sample preparation; however, XL-MS can capture more of these regions. Indeed, mono-links were identified in IDRs of Pgp, K31 before the first elbow helix or K645 and K685 in the linker region of the Pgp, which are all missing from the cryo-EM structure 6qex [13]. A cross-link between K536 and K685 was also identified, which could support determining orientations of the linker [45], since predictions of this region have been controversial so far [46]. The linker region, containing several conserved phosphorylation sites and a high-affinity tubulin binding motif, is probably involved in the transport cycle as well [47]. The K536 near the C-loop of NBS2 in contact with the linker peptide supports the idea of communication between the linker and NBD-s and that the linker regulates ATP hydrolysis and substrate specificity [48].

The K826 has been demonstrated to be in electrostatic interaction with the phosphate group of phospholipids by means of molecular docking and ion mobility mass spectrometry (IMS) approaches [49]. In our experiments, we found K826 to be cross-linked with K786 (Figure 6C inlet, Figure 7A–D) and K808 in the ICH3 loop of Pgp (Figure 6C), which rests on the top of the NBD2 and has been described to become protected from deuteration in the presence of cholesterol in nanodiscs [8]. Moreover, K808 was in a cross-link with K536 at the C-loop of NBD1. In the vicinity of K536, the Walker A region on NBD1 and ICH4, which were also modified by cross-linkers (Figure 6D), has been reported to interact in the presence of lipids [6]. ICH4, which rests on the top of the NBD1, is connected to ECL5 and ECL6 and was reported to become more solvent-accessible upon adding cholesterol to the lipid bilayer, contrarily to ICH3, which became more protected [8]. Cross-links between these regions of the protein support the idea that there is a complex interplay between cholesterols and membrane composition and between Pgp’s structure and function. ECL5 and ECL6 surrounded by cholesterols [13] are directly linked to ICH3 and ICH4, and the asymmetric movements of these helices in the presence of cholesterol raise the possibility that the ECLs change their orientations too (Figure 6E).

Interestingly, K1220, closely connected to the D-loop, which has similarly been described as a cholesterol-dependent region by Clouser et al., was found to be cross-linked with K1150, which was one of the most readily accessible lysine residues for the cross-linkers. All other cross-links found in this study have also been described to be able to become deuterated in HDX experiments [4].

In the living cell experiments, a native lipid environment is well-maintained for Pgp; however, cross-linkers could be hydrolyzed more quickly until they reach lysines on the protein. Therefore, less information can be gained. The on-bead cross-linking method is less physiological, although a zwitterionic detergent was used for membrane preparation, which keeps some of the lipid molecules around Pgp; thus, the protein can maintain its ATPase activity [50]. Shared mono-linked and cross-linked sites between living cell and on-bead cross-linking approaches also support the idea that on-bead experiments are indeed suitable for membrane protein structure mapping. Our results show how protein crystallography and mass spectrometry (XL-MS, HDX, and IMS) can complement each other’s results even with such complex data created from the processing of a membrane protein.

**Figure 5 ijms-24-10627-f005:**
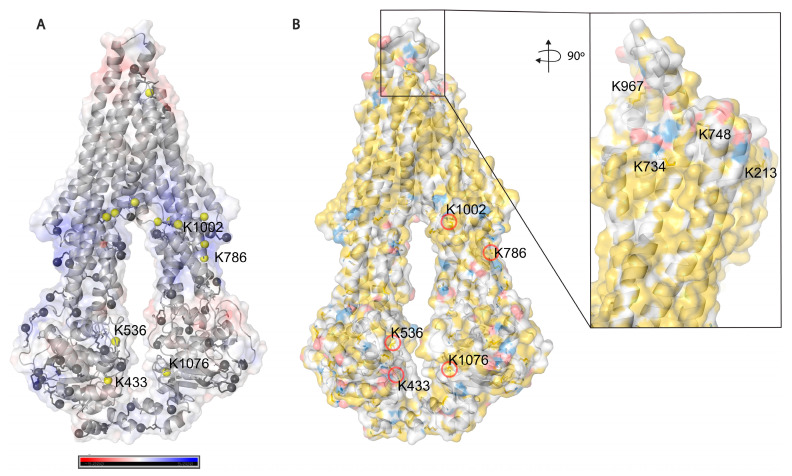
Solvent accessibility of lysines on Pgp. (**A**) Electrostatic potential surface representation of inward-facing Pgp (6qex); nitrogen atoms of all lysines are represented as grey balls, and buried ones are colored in yellow. Five lysines with a legend are the ones which were modified by cross-linkers. (**B**) Hydrophobicity and charge distribution on Pgp’s surface. Carbon atoms not bound to nitrogen or oxygen atoms are yellow, oxygens with negative charges in glutamate and aspartate are red and nitrogens carrying positive charges in lysine and arginine are blue, while all other atoms are white [51]. The inlet shows extracellular lysines of which only K734 and K967 are surface exposed; however, when Pgp is in complex with UIC2 or 15D3 they are also buried.

**Figure 6 ijms-24-10627-f006:**
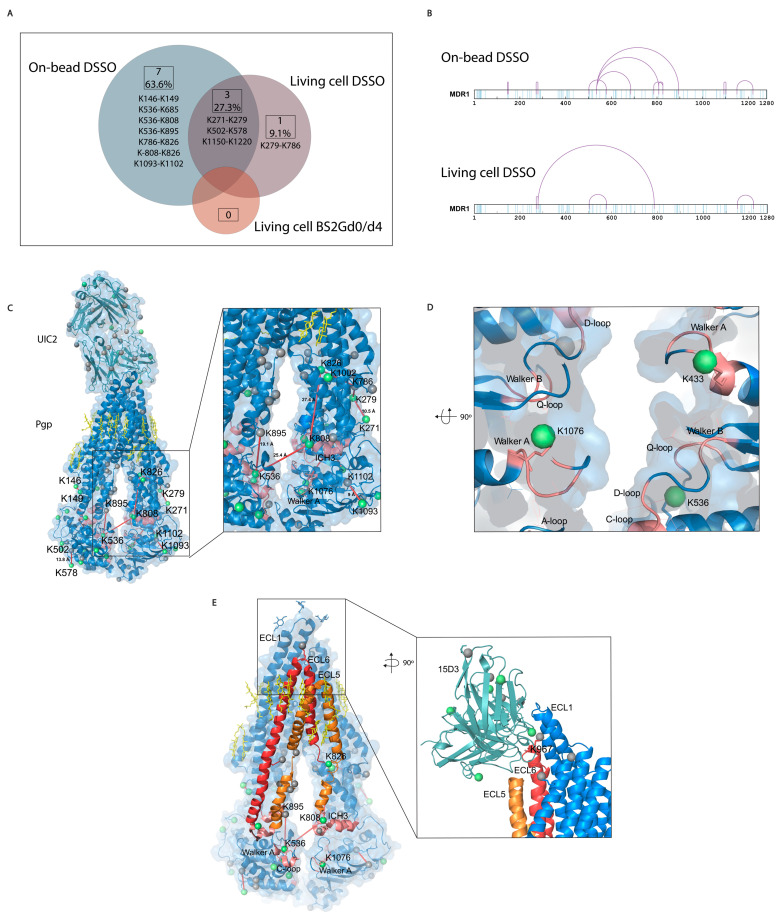
Mono-links and cross-links on Pgp. (**A**) Overlap of cross-links between different sample preparation approaches. Eleven cross-links were identified with all sample preparation approaches, with a 27% overlap between the living cell and the on-bead DSSO methods. No cross-links were identified with the living cell BS2Gd_0_/d_4_ approach. (**B**) Linear plots of cross-linked sites on human Pgp identified with on-bead and living cell approaches. All lysines are indicated in light blue, and cross-links are purple. (**C**) Mono-links highlighted in green balls and cross-links indicated by red lines on the structure of Pgp in complex with UIC2 antibody (6qex) based on the results of the cross-linking approaches. Lysines which are not mono-linked are depicted as grey balls, and cholesterols and lipids are yellow. The inlet shows the ICH3 loop involved in cross-links highlighted in pink, which was previously described to be influenced by the presence of cholesterol. (**D**) Mono-links of Walker A regions and other segments of nucleotide binding sites (NBS) emphasized in pink. (**E**) ICH3 directly connected to ECL5, cross-linked to C-loop, which has cross-link with ICH4, directly connected to ECL6. Connections of these regions on Pgp suggest a complex interplay between segments that have previously been described as cholesterol-sensitive. The inlet highlights the cholesterol-sensitive binding sites of 15D3 mAb, and mono-linked lysines are colored in green. One mono-link on 15D3 fell in the binding region close to Pgp’s K967 lysine on ECL6; however, cross-link formation between them could not be detected unambiguously.

**Figure 7 ijms-24-10627-f007:**
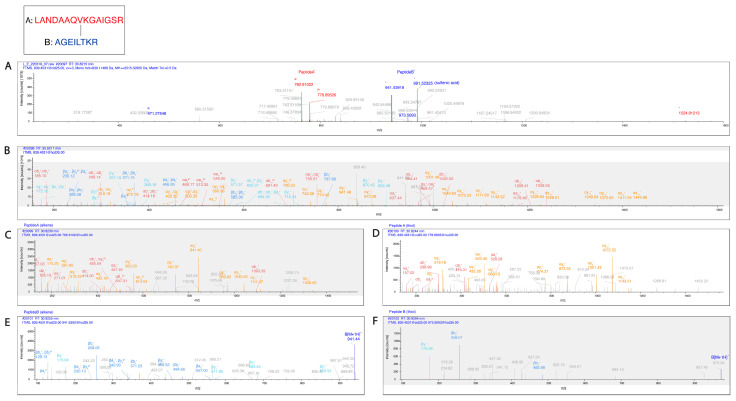
Pgp K786 is cross-linked to K826. (**A**) The MS2-CID spectrum dominated by the diagnostic peptide pairs of the two cross-linked peptides formed upon fragmentation along the cross-linker. (**B**) The MS2-HCD spectrum containing additional fragments of the two peptides along the peptide backbone to assist identification. Fragments labeled red and orange belong to Peptide A while fragments labeled blue and light blue belong to Peptide B. (**C**–**F**) MS3-HCD spectra of the peptide pairs detected in MS2-CID confirm the identity of Peptide A (**C**,**D**) and Peptide B (**E**,**F**). Peptide A is LANDAAQV**K**GAIGSR [818–832], and Peptide B is AGEILT**K**R [780–787]. All other cross-link spectra are available in the Appendix A.

### 2.4. Molecular Docking Suggests ECL1, ECL5, and ECL6 of Pgp for the 15D3 Epitope

The observed cross-links related to the cholesterol-affected NBD connections and the lack of detecting cross-links between the ECLs of Pgp and the applied Abs forced us to model these protein interactions. Since no cross-links were identified between Pgp and the antibodies, this information could not be used for molecular docking. The 15D3 antibody variable heavy and light chain sequences (Patent US 5849877) and the human MDR1 PDB structure 6qex [13] were used for computational analysis.

Based on these docking calculations performed using ClusPro 2.0, the CDRs of the ABodyBuilder2-predicted structure of 15D3 become close primarily to the first and sixth extracellular loops (ECL1 and ECL6) on the substrate-bound human Pgp (Figure 8A). As mentioned above in the introduction, 15D3 binds to Pgp independent of Pgp’s catalytic state; thus, 15D3 can be expected to bind to ECLs which do not move far away from each other under the action of Pgp’s pumping mechanism. ECL1 and ECL6 indeed stay in proximity with each other during the catalytic cycle. Furthermore, a putative binding site on ECL1 could be partially overlapping with the epitope of UIC2 (Figure 8B), which could explain the competition between UIC2 and 15D3 [14].

The assumption of the cholesterol-modulated ICH-NBD rearrangements and the ECL5 or ECL6 movements is corroborated by the docking predictions of 15D3 cholesterol-dependent mAb, which connects to Pgp from the side of ECL5 and ECL6, reaching ECL1 as well (Figure 6E inlet and Figure 8A)

In the case of cross-linking in the presence of cholesterol (living cell method), in proportion, 50% (two out of four: K502–K578 and K1150–K1220) fixation fell on NBDs. On the other hand, in the case of the on-bead samples, where the cholesterol level was decreased, cross-links were concentrated on the linker and ICH3 and ICH4, in 50% (5 out of 10, K536–K685, K536–K808, K536–K895, K786–K826, and K808–K826), which in this case were asymmetrically bound to the roof of NBD1.

Considering mono-links, only two mono-links appear in living cell samples for the linker-ICH3-ICH4 regions. Although these mono-links also appear in the on-bead pattern, three more mono-links (K645, K685, K626) and five more cross-links (see above) appear in this region of the Pgp in these samples than in the case of the living cells. These differences between the two sample preparation approaches suggest a significant rearrangement of the structure in the absence of cholesterol. The change in the distribution of mono-links in the cholesterol-depleted samples suggests that the linker-ICH3-ICH4 connection and its surrounding regions may move in the presence of cholesterol. Similar motional shifts were detected in the HDX experiments, where compaction–decompaction was observed for cholesterol in the ICH3–ICH4 loops, as mentioned above [8].

However, mono-link distribution in other areas of Pgp was not substantially rearranged in living cell or on-bead preparations. This result is an internal control of our observation, as a more intense random behavior of mono-links compared to that of cross-links is to be expected.

This suggests that a vertical rearrangement occurs in the absence of cholesterol, which exactly affects the linker, ICH3 and ICH4 cytoplasmic sequences, attached directly to the ECL4, ECL5 and ECL6 through rigid transmembrane helices. This rearrangement can pull the ECL5 or ECL6 loops vertically into the plane of the membrane, while in the presence of cholesterol, the same loops can protrude more from the membrane. The ECL4 extracellular loop is longer than the ECL5 and ECL6 one, and thus its relative motion may be less significant. Cholesterol may exert its effect through cholesterol-binding motifs, concentrated in excess in the nearby ECL4, ECL5 and ECL6, through a mechanism similar to that of surface tension. In cholesterol abundance, cholesterol can crawl up the “wall” of the cholesterol-binding motifs containing protein, which can deform the horizontal plane of the transporter protein in the level of the membrane. This mechanism is a possible explanation for the cholesterol-dependent behavior of the 15D3 antibody.

The idea of a relation between ECL6 and ICH–NBD interfaces has also been suggested by another molecular dynamics simulation where the mutation of the modulator binding site (M-site) (F978A) near ECL6 decreased the number of contacts between ICH3 and NBD2, but increased contacts at the ICH4–NBD1 interface [2].

### 2.5. Cross-Linking and Affinity Purification MS Are Suitable for Identifying Protein Interaction Partners of Pgp

In the present study, several proteins that were in proximity with Pgp or showed alterations upon Pgp expression modulations were identified via MS. Although the experiments were carried out on mouse fibroblast cells overexpressed with human Pgp, connections between Pgp and its partners can be explained by their conserved sequence among species at their interaction sites. No cross-links were identified between Pgp and other proteins; it can, however, be assumed that after nine intense washes with RIPA buffer, which follows the immunoprecipitation step, most of the contaminants and weak interaction partners are washed off. This is hinted by the overall low occurrence of albumin that was used for blocking the Protein G magnetic beads before immunoprecipitation (Appendix A). In total, 368 proteins were identified with 15D3 mAb, and 222 were identified with UIC2 mAb, of which 208 were found with both mAbs (Figure 9A). A reason why 15D3 could enrich a more versatile set of proteins could be that Pgp interacts with different proteins in its different conformations; thus, 15D3 being able to bind not only to the inward-facing conformations, such as UIC2, but independent of the catalytic cycle, might capture more partners.

Human orthologs of interaction partners in the on-bead samples are visualized as an interactome network based on the STRING database in Figure 9B,C. A semiquantitative abundance of individual proteins is depicted by the size of the nodes based on normalized spectral counts (SPCs). Network figures and lists of the protein identifications from the living cell experimental approaches are in the Appendix A. The high abundance of cytoskeletal proteins in all samples supports the idea of Pgp’s strong connection to the cytoskeleton (Figure 9D–F). Actin [52,53], tubulins [54], and filamin A [55] have previously been described to possess a direct or indirect connection with Pgp. AHNAK, which was identified in all samples with a high number of SPCs, had been shown to have altered expression levels in drug-resistant cells where Pgp was overexpressed [55]. Moreover, heat shock proteins Hsp90 and Hsp70, which were also found in all our approaches, are also known to be associated with MDR and Pgp [56,57]. These chaperones are associated with membrane lipid rafts and cholesterol, and crosstalk between Hsp90 and cholesterol might foster the activity of Pgp [58,59].

Among the 160 uniquely identified proteins of 15D3 enrichment, many belong to the ubiquitin system or caveolae (Figure 9B,E). Different ubiquitin E3 ligases were primarily identified in samples enriched with 15D3 mAb. The most abundant among them was NEDD4 which promotes the ubiquitination and degradation of Pgp [60,61]. Whether or not there is a reason why E3 ligases are more definite in the case of pull-down by the cholesterol-sensitive 15D3 antibody has not been investigated; however, it has been demonstrated that increased cholesterol inhibits the degradation of ABC transporters ABCA1 and ABCG1 via the ubiquitin–proteasome pathway [62]. Additionally, RING finger protein 2, another E3 ubiquitin ligase, was identified with both 15D3 and UIC2 mAbs, which was reported to be in direct connection with Pgp’s linker peptide [63]. In addition, caveolae-associated protein 1 (Cavin-1) identification was also more notable in samples enriched with 15D3 than in those enriched with UIC2. Cavin-1 stabilizes caveolae in a cholesterol-dependent manner [64] by connecting to Caveolin-1 through its scaffolding region in the lateral side of the plasma membrane, enriched with cholesterol and acidic lipids [65]. Caveolae reside in the cholesterol-rich lipid rafts of the plasma membrane, and the increased number of these specific membrane regions promote more aggressive tumor cell growth and MDR [66]. Cavin-1 has an increased expression level in MDR along with Caveolin-1 and Pgp, which was claimed to be necessary for fortifying lipid rafts and MDR [67]. A decent number of Pgps are localized in membrane lipid rafts [68] and in certain cancer cell types; a physical interaction between Pgp and Caveolin-1 and regulation of Pgp by Caveolin-1 was proven [69]. Hinrichs et al. claimed that in other cell types, Pgp and Caveolin-1 had different solubility behaviors in Triton X-100, and Pgp does not dwell in caveolae [70]. Based on our experiments, it can be hypothesized that Pgp has a stronger connection with Cavin-1 and cytoskeletal proteins, and through these connections, Pgp could be regulated by Caveolin-1. Since the binding affinity of the 15D3 antibody is dependent on the presence of cholesterol in the plasma membrane, the enrichment of Pgp by this antibody can pull down more cholesterol-sensitive interactions.

**Figure 9 ijms-24-10627-f009:**
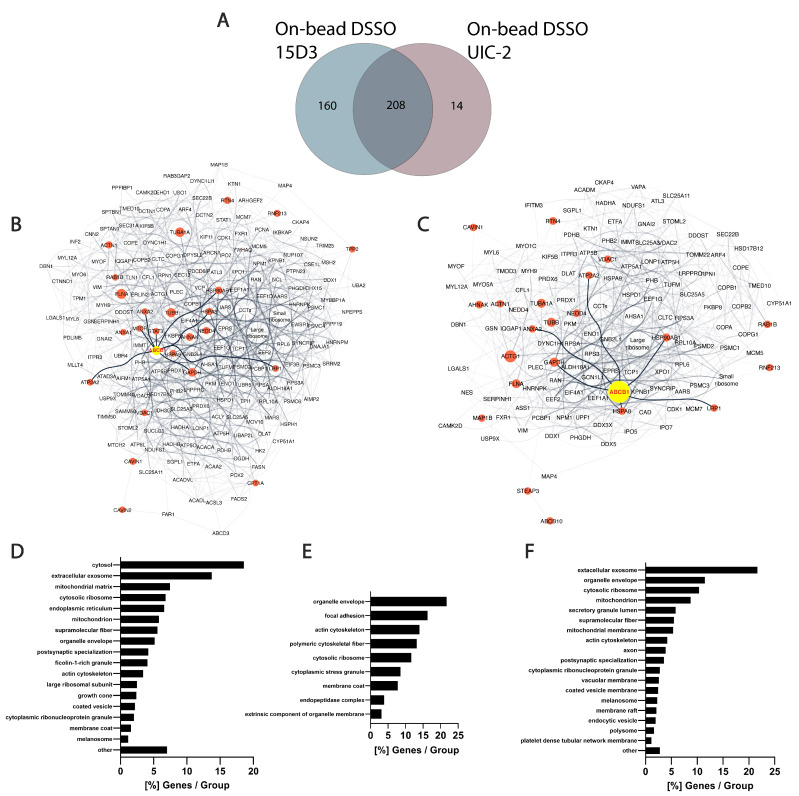
Differences between 15D3 and UIC2 IP enrichments. (**A**) Overlap of protein identifications between the two mAbs. (**B**,**C**) Protein interaction networks of Pgp enriched by 15D3 (**B**) and UIC2 (**C**) mAbs. Interactions were gathered and depicted using the STRING database employing all interaction sources [71] and Cytoscape software accordingly. Only proteins with at least 10 SPCs were considered. Pgp is labeled in yellow, its well-known protein partners and proteins related to MDR are highlighted in red and neighbors of these highlighted nodes are pale grey. The nodes’ size is proportional to the individual proteins’ SPCs. SPCs were normalized to all peptide-spectrum matches (PSMs) in the given analysis. Nodes of the large and small ribosome subunits and components of the chaperonin-containing T-complex were grouped together to reduce the complexity of the network. Pgp was approximately 7 times more abundant based on SPCs using UIC2 for the IP and pulled down 222 proteins, whereas with 15D3 IP a more diverse set of 368 proteins could be enriched, many of which were cholesterol-sensitive. These networks indicate that enriched proteins are indeed related to Pgp. (**D**–**F**) Cellular component distribution of 15D3 enrichment (**D**), the 160 proteins uniquely identified in the 15D3 pull-down experiment (**E**), and proteins enriched in the UIC2 immunoprecipitation experiment (**F**) analyzed with the ClueGO app within Cytoscape [72].

## 3. Materials and Methods

### 3.1. Materials

All chemicals were from Sigma-Aldrich (Munich, Germany) except where indicated otherwise.

### 3.2. Cell Culture

Human *Mdr1* gene-transduced 3T3 mouse fibroblast cells (NIH 3T3 MDR1, a gift from M. Gottesman, stably expressing a high level of cell surface human P-glycoprotein in mouse cells) were cultured at 37 °C in a humidified gas incubator containing 5% CO_2_. Cells were subdivided every second day in Dulbecco’s modified Eagle’s medium (DMEM) supplemented with 10% heat-inactivated fetal calf serum, 2 mM L-glutamine, 25 µg/mL gentamicin, and 670 nM doxorubicin. Two days before the experiments, the culture medium was replaced by a doxorubicin-free culture medium. Cells were regularly tested for mycoplasma.

### 3.3. Antibody Purification

UIC2 (IgG2a; ATCC No. HB-11027) and 15D3 (IgG1; ATCC No. HB-11342) anti-ABCB1 mAbs were prepared from supernatants of hybridoma cells. Supernatants were enriched and purified using Protein G affinity chromatography. Hybridoma cell lines were obtained from American Type Tissue Culture Collections (Manassas, VA, USA). Briefly, 15D3 hybridoma cells were grown on a macrophage feeder layer from BALB/c mouse peritoneal fluid [73] or for selected applications isolated from mouse ascites. Briefly, 15D3 hybridoma cells (1 million cells/mouse) were intra-peritoneally injected into male BALB/c or SCID Swiss nude mice aged 7–11 weeks old using 0.2 mL IFA/mouse (incomplete Freund’s adjuvant). Mice were kept in the Animal Core Facility at the University of Debrecen, housed separately in cages, had ad libitum access to water and chow, and were kept in a 12 h light/dark cycle with a controlled temperature of 22 ± 1 °C. On the 14^th^ day or when required, ascites were collected aseptically and stored at −80 °C. In accordance with the manufacturer’s instructions, the collected ascites were purified using a protein G column kit, Thermo Fisher Scientific (NabTM Protein G Spin Kit, 1ml, Cat No: 89979), to obtain 15D3 mAb. The isolated 15D3 mAb protein concentration was measured at 280 nm using NanoDrop One^C^ microvolume UV-Vis Spectrophotometer (Thermo Fisher Scientific, Budapest, Hungary). If not frozen and stored at −20 °C, Na-azide (from 2% in 100× dilution) was added to the purified antibody and stored at 4 °C.

### 3.4. Flow Cytometry

Briefly, 500,000 cells per sample, previously treated with different concentrations of PNGase F (0, 50, 100, and 200 U/mL) as described below, were placed into sorter tubes in 500 µL of glucose-PBS. Cells were labeled with primary antibodies, 15D3 or UIC2, at a concentration of 10 µg/mL for 30 min at 37 °C. After 3 washes in PBS, cells were labeled with 1 µg/mL Alexa-488 conjugated anti-mouse goat secondary antibody (Thermo Fisher Scientific) for 1 h on ice. Cells were washed 2 times in PBS and resuspended in 100 µL of ice-cold 4% PFA. Cells were analyzed using a NovoCyte flow cytometer (ACEA Biosciences, San Diego, CA, USA). In each sample, 20,000 cells were collected. Cells were gated for single cells according to the FSC-H/SSC-H and SSC-H/SSC-A plots, and then homogenous populations according to the FSC-H/FL1-H dot plot were selected. The FL1-H signal of the Alexa-488 dye was plotted on overlayed histograms. Data analysis and graphs were made using FCS Express version 6 (De Novo Software, Glendale, CA, USA).

### 3.5. Immunoblot Analysis

Cell lysates (7 μg protein/sample) were diluted in SDS sample buffer (0.31 M Tris-HCl, pH 6.8, 50% glycerol, 10% SDS, 100 mM DTT, 0.01% bromophenol blue, and 1 M β-mercaptoethanol) and incubated at 65 °C for 10 min with shaking. Proteins were separated via electrophoresis on 7% SDS-polyacrylamide gel in Laemmli buffer and electro-transferred in Towbin buffer onto a nitrocellulose membrane (Bio-Rad). The membrane was blocked with 5% non-fat dried milk in PBS for 60 min, at RT. Pgp was labeled with D3H1Q (Cell Signaling Technology) human anti-Pgp rabbit monoclonal primary antibody (1:1000 diluted in 5% non-fat dried milk in PBS) for 60 min, at RT with continuous agitation. Non-bound antibodies were washed with PBS containing 0.1% Tween-20 3 times for 10 min, at RT with continuous shaking. They were then labeled with anti-rabbit goat IgG secondary antibody conjugated with horseradish peroxidase (diluted in a ratio of 1:5000 in 5% non-fat dried milk in PBS) for 60 min, at RT with continuous shaking. The unbound antibodies were washed 3 times, then the immunoblots were developed with SuperSignal West Femto ECL reagent (Thermo Fisher Scientific) and images were recorded using the Chemidoc imaging system (Bio-Rad Hungary Ltd., Budapest, Hungary).

### 3.6. Living Cell Cross-Linking Reactions

#### 3.6.1. BS2Gd_0_/d_4_ (Bis(Sulfosuccinimidyl) Glutarate-d0/d4)

NIH 3T3 MDR1 cells were grown in T75 flasks in complete media. Briefly, ~90% confluency cells were trypsinized (0.05% trypsin and 0.02% EDTA in PBS, pH 7.4) for 2 min, and trypsinization was stopped with complete media. Cells were washed twice with serum-free media supplemented with 0.1% molecular-biology-grade BSA to prepare for PNGase F treatment. Furthermore, ~8 × 10^7^ cells/mL in a bacterial Petri dish were treated with 100 U/mL PNGase F (New England Biolabs, Ipswich, UK) diluted with serum-free media supplemented with 0.1% molecular-biology-grade BSA. Cells were incubated for 4 h at 37 °C in a humidified gas incubator containing 5% CO_2_. Cells were collected in 15 mL centrifuge tubes and washed twice with glucose–PBS (phosphate-buffered saline extended with 8 mM glucose). For extracellular antibody cross-linking samples, 10^7^ cells/mL were incubated with 100 µg/mL 15D3 or UIC2 for 30 min at 37 °C with mild agitation. After washing twice with glucose-PBS, at RT (room temperature), 3 × 10^7^ cells per sample were resuspended in PBS, pH 7.0, and BS2Gd_0_/d_4_ (Thermo Fisher Scientific) was added in 1:1 ratio for a final concentration of 5mM dissolved in PBS, pH 7.0, in a final volume of 1 mL. The cross-linking reaction was performed for 60 min at 37 °C, and it was quenched for 5 min at RT with 1 M Tris, pH 7.5, reaching a 20 mM final concentration in the reaction mixture.

#### 3.6.2. DSSO (Disuccinimidyl Sulfoxide)

Living cells were treated with PNGase F as described above and 3 × 10^7^ cells per sample were resuspended in PBS, pH 7.0.; 50 mM DSSO (Thermo Fisher Scientific) stock solution was prepared in DMSO and further diluted with the samples to reach 0.5 mM, 0.5 mM, 1 mM, 5 mM, and 10 mM final concentrations in a final volume of 1 mL. The cross-linking reaction was performed for 60 min at 37 °C, and then samples were treated as described above.

### 3.7. On-Bead Cross-Linking

Reductive methylation [32] of Protein G was performed with sodium cyanoborohydride (NaBH_3_CN) on Dynabeads prior to immunoprecipitation and cross-linking. Briefly, 500 µL of beads were washed three times with PBS, pH 6, then 0.1 M paraformaldehyde (with no methanol), and finally, 0.1 M NaBH_3_CN was added. The mixture was incubated for 30 min, at RT with mild agitation. Beads were washed with PBS, pH 7.4, three times, then placed in a new tube. Membrane preparation and immunoprecipitation with 15D3 or UIC2 antibody were performed as described below. Subsequent to washing with RIPA buffer, beads were washed with PBS, pH 7.0, twice, and the second wash was placed in a new tube. Furthermore, 1 mM DSSO was added to the samples, and had been previously dissolved in DMSO with a 50 mM stock concentration. The cross-linking reaction was carried out for 60 min, at RT with mild agitation, and it was quenched for 5 min with 1 M ammonium hydrogen carbonate (AMBIC), pH 7.5, reaching a 20 mM final concentration in the reaction mixture. Then, beads were washed twice with 25 mM AMBIC (for safety and environmental considerations, these experiments were carried out under a fume hood, and all supernatants and remaining solvents containing NaBH_3_CN were neutralized with sodium hypochlorite.)

### 3.8. Membrane Preparation and Immunoprecipitation Enrichment

Membrane preparation was performed in accordance with the manual of Thermo Fisher Scientific, for Mem-PER™ Plus Membrane Protein Extraction Kit (#89842). For immunoprecipitation, 100 µL of Dynabeads Protein G (Thermo Fisher Scientific) was washed four times in PBS supplemented with 5 mg/mL BSA (PBS/BSA). Cell lysates with 500 µg of protein of a 1 mg/mL concentration were added to the beads and rotated at 4 °C overnight. The following day, the beads were washed nine times in 1 mL of RIPA buffer [23] at 4 °C. Then, the beads were washed twice in 1 mL of freshly made ice-cold 100 mM AMBIC, and the second wash was placed in a new tube. Beads were frozen at this point and stored at −80 °C until trypsin digestion and LC-MS analysis.

### 3.9. Trypsin Digestion and LC-MS Analysis

Beads were resuspended in 25 mM AMBIC, and protein disulfide bonds were reduced with 1 mM tris(2-carboxyethyl)phosphine (TCEP) for 10 min at room temperature. Resulting sulfhydryls were derivatized with 2.4 mM S-methyl methanethiosulfonate (MMTS) for 10 min at room temperature. Protein digestion with side-chain-protected porcine trypsin (Promega Corporation, Madison, WI, USA) proceeded overnight at 37 °C; then, it was terminated by acidifying the samples to obtain a final 0.2% trifluoroacetic acid (TFA). The digests were analyzed using a LC-MS/MS on an Orbitrap Fusion Lumos Tribrid (Thermo Fisher Scientific) mass spectrometer on-line coupled to an ACQUITY UPLC M-Class system (Waters Corporation, Milford, UK) or an Evosep One HPLC system (Evosep). In the ACQUITY UPLC M-Class setup, samples were loaded onto a Symmetry C18 Trap column (100 Å; 5 μm; 180 μm × 20 mm; 2D; V/M) with a flow rate of 5 μL/min for 5 min with 1% solvent B, then separated on a Peptide BEH 130 C18 column (130 Å, 1.7 μm, 100 μm × 100 mm) or a Peptide CSH C18 column (130 Å, 1.7 μm, 75 μm × 250 mm) at a flow rate of 400 nL/min with a gradient of solvent B from 5 to 10% in 2 min, from 10 to 35% in 50 min, from 35 to 50% in 15 min, and then up to 90% in 8 min keeping the column temperature at 45 °C. In the Evosep One setup (Evosep Biosystems, Odense, Denmark), samples were loaded onto Evotip Pure tips and separated using the Extended 15 SPD (samples per day) method on EV-1106 analytical column (C18; 1.9 μm; 150 μm × 150 mm). In both setups, solvent A was 0.1% formic acid in water, and solvent B was 0.1% formic acid in acetonitrile. MS data acquisition was performed in a data-dependent fashion on multiply charged precursors. Non-cross-linked and BS2G-cross-linked samples were analyzed via MS2-HCD fragmentation only. In DSSO cross-linked samples, targeted MS3-HCD spectra were collected whenever the DSSO fragmentation-specific mass difference was detected in MS2-CID, and here, the data acquisition was complemented with MS2-EThcD and MS2-HCD fragmentation as well. MS1 and MS2-CID data were measured in the orbitrap with a resolution of 60k and 30k, respectively, and MS2-HCD data were measured either in the orbitrap with a resolution of 15k or in the ion trap with a rapid scan rate, whereas MS3-HCD and MS2-EThcD data were measured in the ion trap with a rapid scan rate.

### 3.10. Data Processing and Interpretation

Peak lists from raw data were generated using Proteome Discoverer (v1.4 or v2.4) and then submitted for a database search via ProteinProspector (v5.24.0, v6.2.1, or v6.3.1). As a first stage, data were searched against mouse entries in the Uniprot 2017.11.01 database (83,889 entries) or the SwissProt 2021.06.18 database (17089 entries) to identify proteins present in the sample. Sequences of human Pgp, Protein G, the known stretches of UIC2 and 15D3 antibodies, and common contaminants (e.g., human keratins or bovine serum proteins) were also considered. Precursor mass tolerance was set to 5 ppm and fragment mass tolerance was set to 20 ppm (orbitrap data) or 0.6 Da (ion trap data). Only fully tryptic peptides were considered, with a maximum of two missed cleavages. Methylthio modification of Cys residues was used as a fixed modification and Met oxidation, acetylation of protein N-termini, pyroglutamic acid formation of peptide N-terminal Gln residues, and hydrolyzed DSSO or BS2Gd_0_/d_4_ derivatives on lysine residues were used as variable modifications. Protein hits were accepted with a protein/peptide FDR of < 1% and at least two unique peptide identifications per protein. For the semiquantitative analysis of identified proteins, SPCs were normalized to all PSMs in the given analysis. In the second stage, proteins confidently identified with at least 2 or 5 unique peptides (cross-linked in living cells or on-bead, respectively) were considered to identify cross-links between lysine residues, either with DSSO or BS2G, using Protein Prospector or the XlinkX node in Proteome Discoverer (v3.0.1.27) [74]. In the DSSO cross-linked samples, the alkene, unsaturated thiol, and sulfenic acid derivatives (resulting upon fragmentation) were considered for MS3-HCD data as variable modifications on uncleaved lysine residues. Eventually, MS3 data were paired with their respective MS2 data, and cross-links were also manually evaluated for Pgp. Data were deposited in the MassIVE repository (MSV000091991), and proteins and cross-linked peptide lists are available in Appendix A.

### 3.11. Molecular Docking Analysis

The 15D3 antibody structure was modeled using ABodyBuilder2 (RMSD for CDR-H3 of 2.81 Å) the web application using the 15D3 variable heavy- and light-chain sequence from United States Patent US5849877. RMS prediction errors below the 5 Å threshold were computed using the deep learning algorithm of ImmuneBuilder [75]. Prediction errors for each CDR residue are listed in Table 1.

Protein–protein docking was performed using the ClusPro 2.0 [76] web server in the Antibody mode and with automatic non-CDR masking. The PDB structure of 15D3 previously created with ABodyBuilder2 and the human MDR1 PDB structure 6qex was uploaded together with a masking PDB file created in PyMOL for using only extracellular loops of human MDR1 for docking. The best fit, cluster 0, with 177 members, was used for further interpretations via PyMOL.

## 4. Conclusions

Our experimental approaches provide new insights into the structure and protein interactions of Pgp. Mono-links found intracellularly on Pgp’s hydrophilic surface may support investigations regarding NBS assembly and how the linker is involved in the catalytic cycle. Cross-links supplement the information on solvent accessibility with a distance constraint, thus making it possible to detect connections between NBDs, ICH–NBD interfaces, ECLs, and the linker. The cholesterol-associated ECL5 and ECL6 may communicate with ICH3–NBD2 and ICH4–NBD1 interfaces, affecting drug binding and ATP hydrolysis. Identified interaction partners of Pgp highlight Pgp’s strong connections to cytoskeletal proteins as well as the interplay between cholesterol and other MDR-related proteins. The application of the XL-MS technique with the approaches described in this study may support endeavors in understanding the functions of Pgp and other membrane proteins.

## Figures and Tables

**Figure 2 ijms-24-10627-f002:**
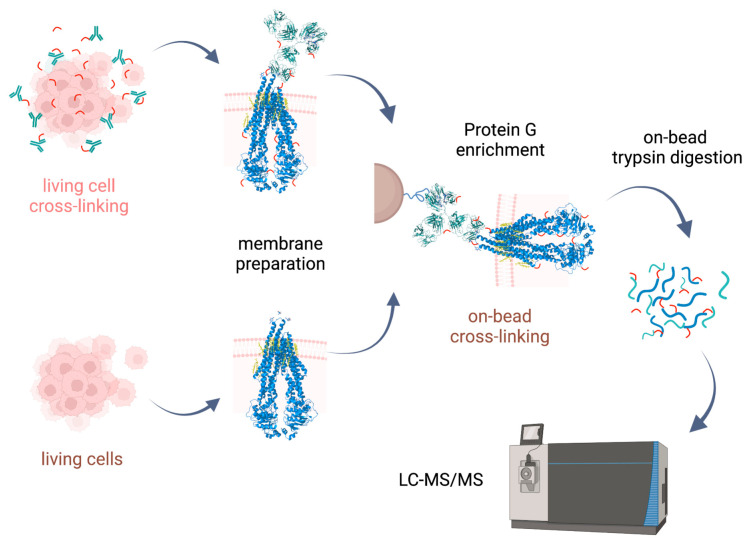
Experimental approaches for XL-MS on Pgp. Cross-linking with lysine-reactive cross-linkers (DSSO and BS2Gd_0_/d_4_) was carried out either on living cells followed by membrane preparation, or on-bead after membrane preparation, then affinity purification. Cross-linked complexes containing Pgp were enriched via extracellular mAbs (15D3 and UIC2). Peptide generation via trypsin digestion was performed on-bead, and measurement of samples was realized via LC-MS/MS. Steps involved only in the living cell approach are indicated in pink, and the on-bead experimental steps are brown, while steps applied in both approaches are indicated in black.

**Figure 3 ijms-24-10627-f003:**
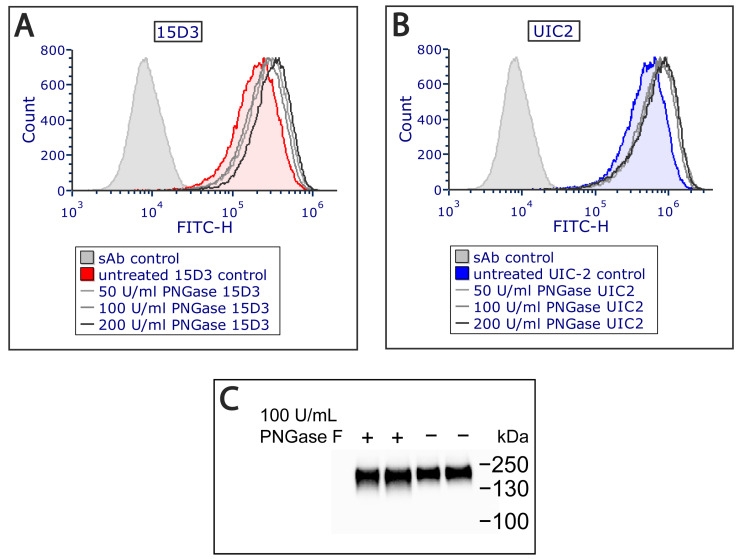
The binding of the (**A**) 15D3 and (**B**) UIC-2 mAbs increased after PNGase F treatment applied at different concentrations on living cells as measured via flow cytometry. N-glycosyl groups sterically affect the docking of both antibodies to their binding site. (**C**) Western blot analysis indicates a small shift in the electrophoretic mobility of the Pgp protein after deglycosylation.

**Figure 4 ijms-24-10627-f004:**
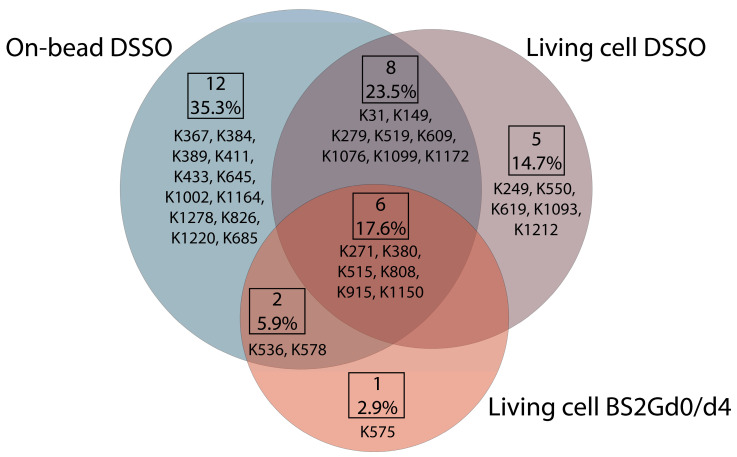
Overlap of unique mono-links on P-glycoprotein in different cross-linking setups. Each type of sample preparation was performed employing UIC2 and 15D3 monoclonal antibodies. Out of 34 unique mono-links, 6 (17.6%) were found in all sample preparation approaches; these are the ones that are most readily solvent accessible. In total, 16 (8 + 6 + 2, 47%) of the mono-links were identified at least with two different sample preparation modes which suggests that, even with distinct methods, it is possible to identify a similar set of modifications. Mono-links detected via each experimental approach visualized on Pgp’s primary structure using Protter 1.0 and on Pgp’s tertiary 6qex PDB structure can be found in Appendix A accordingly.

**Figure 8 ijms-24-10627-f008:**
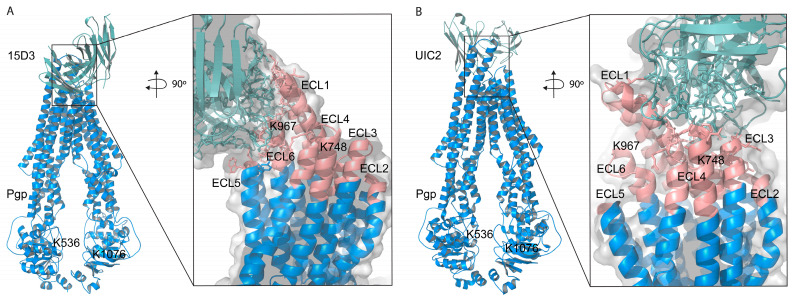
ClusPro 2.0 prediction of docking (**A**) 15D3 mAb to Pgp (6qex); the inlet indicates binding to ECL1, ECL5 and ECL6, partially overlapping with the epitopes of (**B**) UIC2 (ECL1, ECL3, and ECL4), visualized by means of the UIC2-associated, 6qex PDB structure. ECLs of Pgp are highlighted in pink. NHS cross-linkers could potentially target K748 on ECL4 and K967 on ECL6; however, we did not see any modifications in these lysine residues, probably due to the vicinity of the membrane and the antibodies embedding the lysine side chains.

**Table 1 ijms-24-10627-t001:** Prediction errors for each CDR residue of 15D3 mAb.

CDR Residue	Prediction Errors
Framework H-chain	0.33
CDR-H1	0.33
CDR-H2	0.20
CDR-H3	0.20
Framework L-chain	0.20
CDR-L1	0.31
CDR-L2	0.16
CDR-L3	0.18

## Data Availability

Cross-linking MS data are available at the following link: ftp://massive.ucsd.edu/MSV000091991/ (accessed on 19 May 2023).

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
