# Peer review of "Cross-Linking Mass Spectrometry on P-Glycoprotein"

_ijms, 2023, doi:10.3390/ijms241310627_

Round 1
Reviewer 1 Report
The authors are interested in the ABC transporter P-glycoprotein (Pgp) and its role in multidrug resistance in tumor cells. The Pgp protein is found at barrier regions and prevents transport to toxic entities. The Pgp protein contains 12 trans-membrane helices and is expected to crosstalk between the lipid environment and the protein. The two antibodies discussed in detail are 15D3 and the conformational sensitive UIC2 monoclonal, which is selective to the PgP OF state. The 15D3 binds indiscriminately to any PgP on the cell surface (Pool1+Pool2). The authors' primary goal was to study these interactions in near-physiological conditions while maintaining a native lipid environment focusing on the epitopes of the antibody 15D3. The commonly used crosslinking agents are amin-reactive NHS esters which bind to exposed lysine residues. The PgP protein contains 85 lysines, with 70 being surface exposed and 16 buried. It is expected that of the 70 surface proteins, 34 could be chemically labeled, and only 5 of the 16 buried proteins could be chemically modified.
One strength is the authors take great care in their conditions, ensuring the native tertiary structure of the Pgp protein and avoiding the use of formaldehyde which is commonly used in these types of experiments. Instead, the authors successfully used the NHS ester, which is very selective in crosslinking targets. Furthermore, their theory of the 15D3 antibody being selective to a lysine-containing region is supported by PNGase, which selectively cleaves N-linked oligosaccharides in the ECL1 region and has no change in the Pgp activity. Furthermore, mono labeling of the protein indicated that six lysines were solvent accessible, essential for 15D3 binding. The result is strengthened by observing the identical residues being deuterated in previous HDX experiments. The authors also have an excellent experimental setup using DSSO, which can indicate both crosslink and mono-linked C-S bonds.
One minor issue with the assay is that Protein G had to be derivatized to prevent crosslinking events. The need for a modification could indicate that the system might not be suitable to the Pgp protein and could significantly change the binding character of the beads. Another issue is that the authors treat Pgp with the enzyme PNGase. While there is the support that the activity is maintained in prior literature, it would strengthen the manuscript to evaluate it under your conditions.
Overall, the manuscript thoroughly identifies Pgp binding by using crosslinkers that are selective towards lysine groups. Their results are supported not only by previous experiments done independently in other research groups but also computationally. While the results may not be novel the use of lysine-specific crosslinkers to study a membrane protein is an exciting approach that could help future scientists study such complex proteins and provides further evidence of the importance of the lysine-exposed residues.
Reviewer 2 Report
The authors have presented two approaches to perform cross-linking mass spectrometry on an important membrane protein, P-glycoprotein (Pgp), also called MDR1. The two approaches: 1) live cell cross-linking and 2) on-bead cross-linking are reported with improvements for optimal membrane protein cross-linking, including derivatization of protein G beads to improve sensitivity for cross-linked peptides and PNGase-F treatment for improved antibody pull-down. Three cross-linking experiments were performed: 1) On-bead DSSO, 2) Living Cell DSSO, 3) Living Cell BS2G, and three different analysis were performed based on mono-links (Figure 4, Figure 5, Figure 6), Intra-protein cross-links (Figure 6, Figure 7, Figure 8) and inter-protein cross-links (Figure 9).
The approach described is fascinating and the findings presented are potentially interesting to a wider audience interested in MDR1 and its relation to multi-drug resistance. However, there seems to be a small number of cross-links identified which suggests that the cross-linking experiments performed are not adequately optimized. More can also be done to rigorously compare the data obtained based on the different approaches and better present the data in the manuscript.
Questions and Comments:
1. It is difficult to get a sense of the number and coverage of cross-links obtained across the Pgp sequence and how this differs between the three different approaches. The number of Pgp cross-links identified seems to be low (<20 unique cross-links in Figure 6A, 0 cross-links in the living cell BS2G condition), which may indicate that the cross-linking protocols performed are not adequately optimized. Can the authors summarize and state clearly the total number of intra-links (intra-protein cross-links), inter-links (inter-protein cross-links) and mono-links identified from each approach? Provide a discussion on whether the cross-link depth/coverage is sufficient or how it compares to other similar types of cross-linking studies. A circular plot of these cross-linked sites may help better visualize the depth and coverage of the cross-link data reported.
2. Can the authors clarify and state clearly in the manuscript if they are expressing human Pgp in the mouse 3T3 cell lines? And that the human Pgp is being used for their cross-linking experiments?
3. It will be nice to see a more rigorous comparison of the cross-link data obtained from on-bead vs. living cell methods. For example, in Figure 4, which set of mono-links from on-bead or living cell preparation have a better fit with previously published HDX results? In Figure 5, it will be good to mark out the mono-links identified from on-bead and from living cell preparation. A similar comparative analysis should be performed for the cross-link data presented in Figure 6.
4. For cross-links mapped onto the Pgp crystal structure in Figure 6, what is the Cα-Cα distance of these cross-links? Do the identified cross-links match the crystal structure?
5. For the molecular docking analysis, were cross-link data used to guide the docking? If yes, can this be stated clearly and how the cross-link data was used for molecular docking? Are there cross-links identified between Pgp and the 15D3 and UIC2 antibodies? Do the docked structural models agree with the mono-link data? Perhaps marking out or colouring the mono-link and cross-linked residues on Figure 8 will help readers better visualize the cross-linking data on the docked structures.
6. For Figure 9, Figure 9D and 9E are too small to see any meaningful protein names or protein interactions since the font is too small. Can these figures be resized or tweaked to reflect important protein interactors of Pgp and their relative abundance by spectral counts? Have the authors considered using emPAI to normalize their spectral count quantification? Can the authors comment on the 160 proteins identified in 15D3 sample but not UIC-2 sample, maybe perform a pathway analysis on these 160 proteins as done in Figure 9B and 9C?
Round 2
Reviewer 2 Report
Authors have sufficiently addressed the reviewer's comments and questions with the additional comparison to literature sources, improvements to data presentation and additional analysis provided.